# Multilingual Spoken Words Corpus

**Mark Mazumder**  
Harvard University

**Sharad Chitlangia**  
Harvard University

**Colby Banbury**  
Harvard University

**Yiping Kang**  
University of Michigan

**Juan Ciro**  
Factored/MLCommons

**Keith Achorn**  
Intel

**Daniel Galvez**  
NVIDIA

**Mark Sabini**  
Landing AI

**Peter Mattson**  
Google

**David Kanter**  
MLCommons

**Greg Diamos**  
Landing AI

**Pete Warden**  
Google

**Josh Meyer**  
Coqui

**Vijay Janapa Reddi**  
Harvard University

## Abstract

Multilingual Spoken Words Corpus is a large and growing audio dataset of spoken words in 50 languages collectively spoken by over 5 billion people, for academic research and commercial applications in keyword spotting and spoken term search, licensed under CC-BY 4.0. The dataset contains more than 340,000 keywords, totaling 23.4 million 1-second spoken examples (over 6,000 hours). The dataset has many use cases, ranging from voice-enabled consumer devices to call center automation. We generate this dataset by applying forced alignment on crowd-sourced sentence-level audio to produce per-word timing estimates for extraction. All alignments are included in the dataset. We provide a detailed analysis of the contents of the data and contribute methods for detecting potential outliers. We report baseline accuracy metrics on keyword spotting models trained from our dataset compared to models trained on a manually-recorded keyword dataset. We conclude with our plans for dataset maintenance, updates, and open-sourced code.

## 1 Introduction

Keyword spotting (KWS) is a core technology for consumer-facing voice-enabled interfaces on endpoint devices and is ubiquitous on smartphones (e.g., Siri and Google Voice Assistant). Keyword spotters are typically used as lightweight wake-word engines, constantly running on-device listening for a single phrase. Upon detection, the device may invoke a larger speech model for voice-based tasks such as turning on a light bulb. In recent years, neural network-based KWS has shown great promise [34, 4, 18, 6, 5]. State of the art models have demonstrated high classification accuracy [4], sufficient for wide-scale deployment of KWS across a range of consumer use cases and consumer-facing devices. However, a dearth of open-source, multilingual, and diverse datasets precludes extending the benefits of KWS across the world, especially for under-represented groups.

Traditionally, datasets for keyword spotting models require significant manual effort to collect and validate thousands of utterances for each keyword of interest. Consequently, much of the literature relies on existing keyword datasets such as Google Speech Commands [32] and Hey Snips [8]. These datasets are usually monolingual and contain only a handful of keywords in controlled low-noise environments. Moreover, KWS must be robust to a wide range of speaker characteristics (e.g. accents, genders, tone, pitch and stress) and environmental settings. This presents an additional level of challenge in sourcing a corpus that is rich and diverse.

In this paper, we automate the generation of a multilingual keyword dataset to ensure reproducibility and maintainability for both academic research and commercial use. We develop the Multilingual

Table 1: Multilingual Spoken Words Corpus (MSWC) compared to existing keyword datasets.

| Dataset | Languages | Keywords | Hours | Commercial Use | Keyword Generation |
|---|---|---|---|---|---|
| **MSWC (This Work)** | **50** | **344,286** | **6601.39** | **Yes** | **Automatic** |
| Hey Snapdragon [14] | 1 | 4 | 1.19 | Yes | Manual |
| Hey Snips [8] | 1 | 1 | 127 | No | Manual |
| Speech Commands [32] | 1 | 35 | 27.92 | Yes | Manual |
| Common Voice Single Target [1] | 34 | 14 | 141 | Yes | Manual |

Spoken Words Corpus (MSWC) dataset by applying forced alignment on Common Voice [1], a crowdsourced speech corpus. Prior work has used forced alignment to automate the extraction of keywords from speech corpora [13], but challenges in validating the quality of the alignments have precluded the establishment of a reference corpus. In addition to our dataset, we provide an outlier metric for each extracted keyword that reflects the quality of the sample. Our contributions include:

- We release a large multilingual keyword dataset containing over 23.4 million 1-second spoken examples for over 340,000 keywords from approximately 115,000 source speakers across 50 different languages, which correspond to over 5 billion speakers across the planet [10].

- We provide forced alignments containing per-word timings for each Common Voice source sentence across all 50 languages.

- To facilitate comparative benchmarks, we split the corpus into `train`, `dev`, and `test` splits for each keyword with an 80:10:10 division.

- We identify anomalous samples via a self-supervised nearest-neighbors strategy. An outlier metric enables a user to select between a larger noisier dataset, and a smaller cleaner dataset.

- We quantify the domain gap between manually recorded and automatically extracted keywords. We establish the relationship between our outlier metric and the ability for a KWS model trained on recorded keywords to correctly recognize their corresponding extractions.

## 2   Related Work

Most keyword spotting datasets tend to be small and often only provide a single language (Table 1). The Speech Commands dataset [32] is the current standard for keyword spotting research. The dataset contains 105,829 one-second utterances. However, it supports only 35 words in English. Other keyword datasets like the Qualcomm Hey Snapdragon Keyword Dataset [14] and the Hey Snips Dataset [8] target specific "wake words," limiting potential applications. The Common Voice Single Target dataset  [1] is the only keyword dataset to support a wide number of languages, but it only contains 14 keywords (digits 0-9 and 4 predefined keywords) per language. Speech synthesis has been investigated for KWS but can only generate training data in high resource languages [33, 16].

The majority of speech datasets focus on sentence length alignment for speech-to-text applications. Large scale, multilingual datasets like Common Voice [1] and Multilingual LibriSpeech [23] provide tens of thousands of hours of speech data in multiple languages. However, keyword spotting applications require individual words for training, and therefore cannot readily take advantage of these large speech corpora. Forced alignment techniques have been used to generate keyword spotting training data from larger sentence length datasets [13, 2, 18]. However, [13, 2] only use English alignments, and [18] only analyzes 440 keywords in 22 languages, whereas our dataset contains over 340,000 keywords in 50 languages. Additionally, we provide a metric to assess the quality of extractions (Sec. 4.6), and we plan to regularly update and grow our dataset (Sec. 7).

## 3   Ontology of Multilingual Spoken Words Corpus

In this section, we introduce our Multilingual Spoken Words Corpus (MSWC) dataset, which contains spoken words from 50 different languages, ranging from high-resource languages such as English and Spanish to low-resource languages such as Oriya (an Indo-Aryan language spoken in the Indian state of Odisha) and Dhivehi (spoken in the Maldives). In total, the MSWC dataset has 23.4 million unique audio clips (Figure 1). The entirety of the MSWC dataset is open-sourced

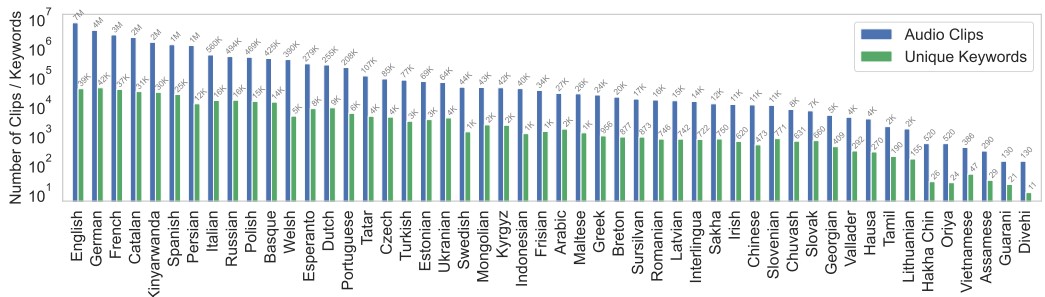

Figure 1: The number of audio clips and unique keywords for each of the 50 languages in MSWC.

(including the forced alignments used to generate our dataset) at `https://mlcommons.org/en/multilingual-spoken-words` along with our code.[1] We also provide a detailed analysis of the MSWC dataset by summarizing its properties via its constituent languages and keywords.

## 3.1 Wide-range of Languages

Our dataset's primary goal is to provide keyword spotting training and testing data for languages with previously limited or non-existent publicly available data, e.g., Italian and Ukrainian with 68 million and 33 million speakers in total respectively. Our dataset contains keyword audio clips for 50 languages. We define the resource level of a language in our dataset by the total number of hours of audio present. Languages with under 10 hours of spoken word samples are referred to as *low-resource*, those between 10 and 100 hours are *medium-resource* and languages with over 100 hours are *high-resource*. The MSWC dataset contains spoken word audio for 26 low-resource languages, 12 medium-resource languages and 12 high-resource languages (Table 2). Out of these 50 languages, the MSWC dataset is, to the best of our knowledge, the only open-source dataset with spoken word data for 46 languages. Download sizes for each subset are also shown in Table 2.

## 3.2 Unique Keywords and Audio Clips

Figure 1 shows the number of unique keywords and audio clips in the MSWC dataset. Our audio clips are opus-compressed using a single channel with a 48KHz sample rate. For high-resource languages, as defined in Table 2, the MSWC dataset has on average 23,408 unique keywords and 1,847,608 audio clips per language. To provide a very coarse estimate of the potential outreach of this subset of our data, we note that these languages are collectively estimated to be spoken by 2.75 billion people worldwide according to Ethnologue [10]. We expect the large swath of keywords and audio samples in MSWC will facilitate KWS models and capabilities for many domains and use cases in these languages. Similarly, for medium-resource languages, the MSWC dataset provides on average 4,054 unique keywords and 109,579 audio clips per language. Several languages, such as Ukrainian (33M people [10]), are getting their first publicly available spoken word dataset.

For low-resource languages, the MSWC dataset contains on average 552 unique keywords and 10,431 audio clips per language. The languages in this low-resource category have the fewest unique keywords and audio clips due to the limited data currently available in Common Voice. Still, we expect this data to enable experiments and exploration into keyword-related tasks in these languages for the first time. Recent work shows that just three to five keyword training examples are sufficient to fine-tune an embedding model for keyword spotting and achieve a high keyword classification accuracy across a wide variety of different languages [13, 18]. Therefore, just a handful of keyword examples can enable unprecedented research and applications for low-resource languages.

## 3.3 Keyword Length Variation

We characterize the relative frequency of word lengths in MSWC by language. Figure 2 shows a distribution of the number of extractions versus the number of characters per keyword for each language. We apply a minimum character length of three as part of the data extraction pipeline as a

---

[1] `https://github.com/harvard-edge/multilingual_kws`.

Table 2: 50 Languages in the MSWC dataset and the size of their corresponding sub-dataset and the number of hours of audio available in MSWC. The languages are organized into 3 resource levels.

| Availability | Languages (Size in MSWC, Hours of Audio in MSWC) |
|---|---|
| Low Resource <10 hours | Arabic (0.1G, 7.6h), Assamese (0.9M, 0.1h), Breton (69M, 5.6h), Chuvash (28M, 2.1h), Chinese (zh-CN) (42M, 3.1h), Dhivehi (0.7M, 0.04h), Frisian (0.1G, 9.6h), Georgian (20M, 1.4h), Guarani (0.7M, 1.3h), Greek (84M, 6.7h), Hakha Chin (26M, 0.1h), Hausa (90M, 1.0h), Interlingua (58M, 4.0h), Irish (38M, 3.2h), Latvian (51M, 4.2h), Lithuanian (21M, 0.46h), Maltese (88M, 7.3h), Oriya (0.7M, 0.1h), Romanian (59M, 4.5h), Sakha (42M, 3.3h), Slovenian (43M, 3.0h), Slovak (31M, 1.9h), Sursilvan (61M, 4.8h), Tamil (8.8M, 0.6h), Vallader (14M, 1.2h), Vietnamese (1.2M, 0.1h) |
| Medium Resource >10 & <100 hours | Czech (0.3G, 24h), Dutch (0.8G, 70h), Estonian (0.2G, 19h), Esperanto (1.3G, 77h), Indonesian (0.1G, 11h), Kyrgyz (0.1G, 12h), Mongolian (0.1G, 12h), Portuguese (0.7G, 58h), Swedish (0.1G, 12h), Tatar (4G, 30h), Turkish (1.3G, 29h), Ukrainian (0.2G, 18h) |
| High Resource >100 hours | Basque (1.7G, 118h), Catalan (8.7G, 615h), English (26G, 1957h), French (9.3G, 754h), German (14G, 1083h), Italian (2.2G, 155h), Kinyarwanda (6.1G, 422h), Persian (4.5G, 327h), Polish (1.8G, 130h), Russian (2.1G, 137h), Spanish (4.9G, 349h), Welsh (4.5G, 108h) |

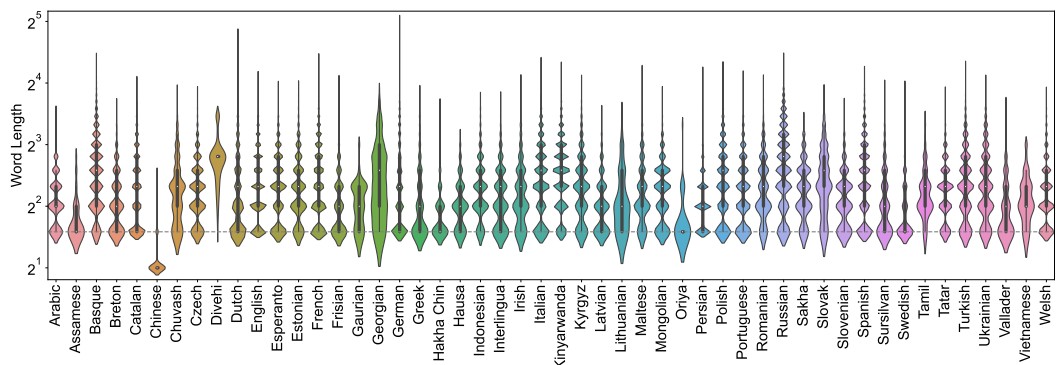

Figure 2: Distribution of word lengths across all 50 languages.

coarse stop-word filter (Sec. 4.4). We note that violin plot smoothing artifacts appear under three characters for some languages. The exception is Chinese where we also extract two character words, as many words in Chinese are two characters; we use the zh-CN language variant. The visualization is normalized within each language and does not show relative distributions across languages.

Figure 2 shows that the keywords in our dataset have a high degree of diversity across all 50 languages. This diversity is more prominent in medium and high-resource languages. For low-resource languages, our dataset contains keywords with the number of characters ranging from three to 19, where there are 619 unique extractions for words with more than 13 characters. Our medium resource language sub-dataset features keywords that range from three to 29 characters long. There are 253 extractions for keywords with >19 characters. Our high-resource subset features keywords that range from three to 34 characters long and 1298 extractions for words longer than 23 characters.

### 3.4 Keyword Characterization

We characterize the parts-of-speech and semantic meanings of keywords in English, Spanish and Arabic, as representative candidates from high-resource and low-resource languages in Table 2.

**Parts-of-Speech** We leverage the "log-linear part-of-speech tagger" [15] in the multilingual Stanza [24] library to assign part-of-speech (POS) tags to keywords in English, Spanish and Arabic. Table 3 shows POS tags for keywords in these languages with the number of corresponding clips and keywords, along with 3 examples (an extended version is provided in the Appendix). We observe the most common categories are nouns, verbs and adjectives, demonstrating our applicability to keyword spotting tasks, as keywords of interest in voice interfaces are often nouns and verbs. Arabic, a low-resource language, contains 236 unique nouns and 102 unique verbs each with more than 2000 extractions. We provide a significant amount of novel keyword data in low-resource languages which we expect to enable new advancements. We also conduct POS tag analyses for six additional

Table 3: Parts of Speech Tags for words in English, Spanish and Arabic. Number of clips (# C) and keywords (# K) in that language. Rows are sorted by the number of keywords in English.

| POSTag | # C | # K | English | # C | # K | Spanish | # C | # K | Arabic |
|---|---|---|---|---|---|---|---|---|---|
| Noun | 2M | 19K | boy, time, man | 491K | 13K | parte, ciudad, años | 2K | 236 | شيع، الكتاب، السيارة |
| Proper noun | 382K | 10K | i'm, i'll, i've | 38K | 725 | del, juan, york | - | - | - |
| Verb | 1M | 6K | are, have, had | 151k | 5k | tiene, encuentra, puede | 2K | 102 | كان، كنت، كانت |
| Adjective | 628K | 4K | other, more, many | 179K | 6K | gran, primera, mismo | 643 | 51 | الكثير، جميلة، القليل |
| Adverb | 683K | 1K | there, also, about | 56K | 378 | más, muy, además | 74 | 2 | كذلك، فقط |
| Pronoun | 644K | 97 | you, that, his | 121k | 117 | los, las, una | 44 | 5 | ارائها، قال، رايته |
| Interjection | 40K | 69 | like, please, well | 2k | 31 | hey, joder,adiós | - | - | - |
| Auxiliary | 305K | 46 | was, can, will | 54k | 98 | fue, son, está | - | - | - |
| Numeral | 121K | 35 | one, two, three | 29k | 42 | dos, tres, cuatro | 217 | 6 | واحد ثلاثة عشر |

Table 4: Semantic keyword characterization in English, Spanish and Arabic via zero-shot multilingual NLI with representative samples. Number of clips (#C) and keywords (#K). Sorted by #K in English.

| Category | # C | # K | English | # C | # K | Spanish | # C | # K | Arabic |
|---|---|---|---|---|---|---|---|---|---|
| Event | 3M | 1K | School, Preakness | 153K | 238 | Campeonato, Episodio | 2K | 30 | قال، حصل |
| Human activity | 1M | 820 | Shooting, Prefers | 78K | 211 | Expandiendo, Visitando | 1K | 28 | استغادر، المشتري |
| Location | 391K | 379 | Home, County | 213K | 324 | Zona, Pueblo | 333 | 14 | ي قال، حصل |
| Name | 327K | 303 | Margot, Cooney | 30K | 128 | Eduardo, Peter | 1K | 24 | اذا ساعدني |
| Animal | 131K | 281 | Sheep, Camel | 99K | 137 | Águila, Especies | 1K | 25 | فيه حافة |
| Number | 301K | 279 | Second, Six | 31K | 75 | Multiplicar, Primero | 493 | 17 | عشرة، واحد |
| City | 106K | 212 | York, London | 34K | 94 | Madrid, Berlín | 49 | 4 | ايطاليا، ديكاس و |
| Technology | 52K | 191 | Television, Videotape | 21K | 106 | Automotriz, Lego | 92 | 5 | التلفاز، السيارات |
| Culture | 52K | 180 | Popularize, Music | 15K | 86 | Tradicional, Concertista | 57 | 4 | الصينية بمهارة |
| Game | 48K | 131 | Play, Kirby's | 8K | 46 | Deportes, Partido | - | - | - |
| History | 52K | 119 | Stories, Ancient | 12K | 73 | Emperador, Históricas | 257 | 8 | قبل اءتذكر |

languages (German, Greek, Russian, Turkish, Vietnamese, and Chinese) in Appendix Fig. 4, and we open-source our code to enable users to similarly explore the other languages in MSWC.

**Semantic Categorization** We inspect keywords in our dataset by semantic class. We apply a Natural Language Inference (NLI) model based on RoBerta [7] on keywords in English, Spanish and Arabic and categorize each word into predefined categories based on common Wikipedia categories (e.g., science, philosophy, religion). Table 4 samples several categories with 2 examples each (expanded further in Appendix Table 10). We observe 60% of keywords belong to *event*, *human activity*, *name* and *animal* categories, and find a relatively even distribution among remaining categories. Appendix Fig. 5 includes six additional languages (German, Greek, Russian, Turkish, Vietnamese, and Chinese), and as above, we provide code to facilitate user analysis of other languages in our dataset.

## 3.5 Speaker Analysis and Background Noise

Our corpus contains an upper bound of 115,000 speakers across 50 languages. The number of speakers for a given language broadly corresponds with the language's resource level. The median speaker count for high, medium, and low-resource languages are 4023, 258, and 54, respectively. Note that the speakers originally contributed their audio samples to Mozilla Common Voice, subject to its collection and validation practices. Speakers had discretion in whether their demographic info was collected during a recording session. If a speaker was not logged in across multiple sessions, they may appear as multiple speakers. In terms of gender diversity, the MSWC corpus contains 60% male, 15% female and 25% unknown speakers.

Common Voice's crowd-sourced recordings are performed in uncontrolled settings on computers and mobile devices, and exhibit background noise (e.g., static, wind, mouse clicks, background voices) which can improve the robustness of speech models in real-world settings. We note that for additional robustness, our dataset can be further augmented with sources of synthetic noise [32] or environmental samples such as babbling, traffic, and crowds [25].

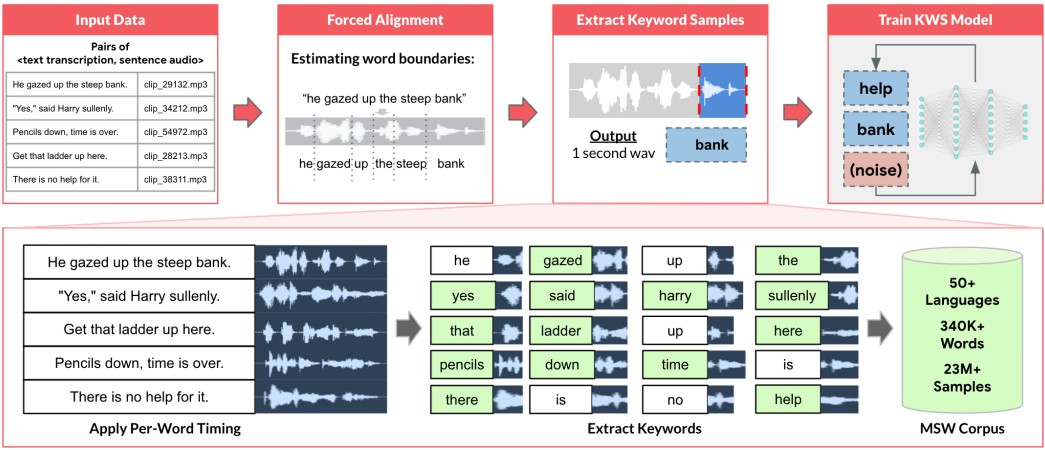

Figure 3: Our automated data extraction pipeline is designed to support multiple languages within the same general workflow, so that we can readily scale to include more than 50 languages over time.

## 4 Constructing the Multilingual Spoken Words Corpus

### 4.1 Data Extraction Pipeline

One of our key goals is to develop an automatic and scalable data extraction pipeline so that we can scale to include more languages and keywords per language over time. To this end, we provide an overview of our automated keyword extraction pipeline and its constituent elements, which we used to generate MSWC. Our pipeline is summarized in Fig. 3. The input to our pipeline is one or more speech datasets consisting of a tuple of ⟨sentence audio, text transcription⟩ pairs, and as output, we produce: (a) a set of word-aligned timing estimates for the dataset, (b) a dataset of extracted keywords, and (c) data splits for training, validation, and testing.

### 4.2 Audio and Transcription Sourcing

We generalize our pipeline so that our dataset can grow to include new languages and additional data in our current languages. Currently, we source all of our sentence audio and transcription data from the Mozilla Common Voice [1] project. But nothing precludes us from using our pipeline to also support data ingestion from a variety of other large speech datasets. Examples of such datasets include The People's Speech [11] and Facebook's Multilingual LibriSpeech [23]. We plan to include these sources as part of our regular updates to the MSWC dataset (Sec. 7).

Common Voice is a large and growing crowd-sourced effort to collect speech data in both widely spoken and low-resource languages. For each language, Common Voice collects public domain text from online sources such as Wikimedia along with user-contributed sentences, and volunteers record themselves reading these sentences through a web-based API. Common Voice provides a validated subset of their crowdsourced data where at least two users have listened to each submitted sentence and affirmed that the spoken audio matches the text transcription. Our initial release of the MSWC dataset utilizes the validated subset of Common Voice version 3. Common Voice encodes each user-recorded sentence as a 48KHz MP3. For each language, we feed all sentence audio and text transcriptions into the next stage of our pipeline.

### 4.3 Forced Alignment

We use Montreal Forced Aligner [19] to generate per-word timing estimates from each pair of audio files and transcriptions. Forced alignment is a well-established family of techniques in speech processing for estimating the occurrence of speech events along multiple boundaries (e.g., syllables, words, or sentences). We train forced alignment from a flat start only on the Common Voice data itself, i.e., we do not rely on any external acoustic models. We use graphemes for our lexicons for each language, and alignment is performed via Baum-Welch for expectation maximization. Since

alignment is a compute-intensive process, requiring multiple days of CPU time for larger languages in Common Voice (e.g., English, German), we include all the generated Praat textgrid alignments as part of our dataset. Each alignment provides per-word timing estimates covering the entire sentence (i.e., not restricted to the keywords we extract), enabling speech researchers and commercial users to immediately leverage our alignments for all 50 languages. Since we cannot manually validate the timing estimates produced by forced alignment, we provide an algorithm to detect outliers in Sec. 4.6.

### 4.4 Word Extraction

Once the word alignments are generated for each audio file, we apply several heuristic thresholds to extract a subset of the words present as individual keyword samples. For 49 of the 50 languages in MSWC, we choose a minimum character length of 3 for performing extractions, excepting Chinese (zh-CN) which uses 2 due to the preponderance of shorter words, as discussed in Sec. 3.3. Since Chinese is non-space-delimited, we estimate word boundaries in text transcriptions via Stanford CoreNLP [17]. We also only extract words which have a minimum of five occurrences in Common Voice. Additionally, we perform text normalization and cleaning where possible (e.g., we filter quotation marks out of transcripts).

Each extraction is encoded in the opus file format using a 48KHz sample rate (the same sample rate as Common Voice), and is stored in a two-level nested subdirectory, where the grandparent directory is the language's ISO code (following Common Voice's convention) and the parent directory is the keyword. The name of each extracted audio file is the same as that of the source Common Voice audio file, enabling users to easily refer back to the source audio file, transcript, and available demographic data using Common Voice's `validated.tsv` metadata file. If the same keyword appears in a Common Voice audio clip multiple times, each extraction is appended with a double-underscore and increasing index number (e.g., `common_voice_de_18122909__2.opus`).

### 4.5 Data Splits

For each language, we provide index files defining `train`, `dev`, and `test` data splits. We anticipate users will commonly operate on a small subset of keywords, and thus split per keyword rather than per language set. Each keyword retains at least one clip in each of the `train`, `dev`, and `test` splits, conforming to a target ratio of 80:10:10 as the clip count increases. Maximizing speaker diversity is prioritized when performing data splits. For each keyword, we first match and group clips by their original Common Voice source's `client_id`. We then ensure that no known speakers appear in multiple data splits for the same keyword. Once a clip is assigned to a split, the clip's keyword and `client_id` pairing is maintained across future releases. Gender balance is also optimized across splits. For keywords with sufficient clip counts, gender balance is prioritized for `test` and `dev` splits.

### 4.6 Estimating the Quality of Extractions via Self-Supervised Anomaly Detection

We empirically identify several (non-exhaustive) sources of error for automated keyword extractions: (1) mismatches in <transcript,audio> pairs from Common Voice, (2) incorrect word boundary estimates from forced alignment, (3) mispronounced words, or (4) loud background noise. While existing speech recognition tools may be capable of detecting outlier samples in some languages, we recognize the need to provide sample quality estimates across all languages in our dataset.

We estimate whether a keyword sample is an outlier via clustering and nearest-neighbors, similar to prior work in self-supervised anomaly detection for computer vision [26, 28, 27]. For a given keyword, we randomly choose a small number of samples (e.g., 50), referred to as the *training set* in the remainder of the section. We construct a feature vector representation for each sample and cluster these features using $k$-means. For the remaining samples, we calculate the euclidean distance of the feature vector representation to each of the cluster centers, and choose the smallest distance as our outlier metric.

| Word | # Clips | Near | Far |
|---|---|---|---|
| may | 7551 | 4% | 32% |
| did | 14353 | 32% | 32% |
| shirt | 1299 | 0% | 54% |
| taken | 1734 | 0% | 12% |
| watch | 1476 | 4% | 28% |
| entire | 1354 | 0% | 12% |
| nature | 1018 | 0% | 12% |
| reading | 1594 | 8% | 58% |
| outside | 2075 | 0% | 2% |
| current | 1836 | 10% | 4% |
| followed | 1307 | 0% | 10% |
| provided | 1627 | 0% | 4% |
| political | 1474 | 0% | 8% |

Table 5: Word error rates (WER) by outlier metric.

For the feature representation, we use an embedding model trained on our dataset [18] which extracts a 1024-dimensional feature vector representation from a 49x40 spectrogram input. We empirically

select $k = 5$ as the number of clusters,[2] and postulate that this is sufficiently large to capture desired speaker diversity in feature space (namely, capturing variance in genders, accents, and age as inliers) while remaining small enough to reduce the likelihood of assigning a cluster to any outliers which may be present in the random training set. The larger the value of the outlier metric, the more likely we estimate the sample to be an error. This enables our dataset users to choose a threshold for discarding $n\%$ of the keyword samples with the largest outlier metric values. We provide users of our dataset with the ability to construct this outlier metric for any sample in our dataset by releasing our anomaly detection code and model.

To assess the efficacy of our outlier metric, we conduct listening tests on 75 randomly chosen keywords across English, Spanish, and German, and report sample results for English in Table 5 (full results in Appendix 1.1). For each keyword, we list the number of extractions in our dataset (# Clips) and among each we choose 50 at random to cluster ($k = 5$) via our feature extractor. We then sort the remaining clips by minimum distance to each cluster in feature space, and report the Word Error Rate (WER) as a percentage when listening to the nearest 50 and farthest 50 samples in the sorted dataset. Table 5 shows a clear trend that the word error rates of the nearest 50 samples for each keyword (with an outlier metric mean $\pm$ stdev value of $1.67 \pm 0.19$) is significantly smaller than the word error rates of the farthest 50 samples (with a correspondingly higher outlier metric mean $\pm$ stdev of $3.96 \pm 0.3$). We also evaluate 25 randomly chosen words each in Spanish and German using the above approach. For the closest 50 samples, we observe an average WER of 4.6% in Spanish and 1.04% in German, whereas for the farthest 50 samples we estimate an average WER of 21.4% in Spanish and 29.6% in German. Our detailed results for Spanish and German are provied in Appendix Table 9.

We note several characteristics of outliers observed in our dataset. Forced alignment is likelier to struggle to accurately capture word boundaries for shorter words (Table 5 and Appendix Table 9 are sorted by character length) and outliers often consist of a portion of a one-syllable word, which leads to higher WER across all distances on short (3 and 4 character) words. Conversely, alignment-based extractions for long, multisyllabic words (e.g., 9 and 10 character words) often remain discernible across all distances, resulting in a lower overall WER where only true outliers are filtered (e.g., missing words in the original recording). Our distance metric is most informative for medium-length words, where errors consist both of true anomalies and misplaced word boundaries from forced alignment, for example with the words *shirt* and *watch*. Lastly, with 50 training samples and 5 clusters, training outliers can occasionally be assigned clusters (e.g., with *did* and *current*), and in practice, users of our dataset may wish to tune these hyperparameters.

## 5    Evaluating Multilingual Spoken Words Corpus

We evaluate the ability to train keyword spotting models using our dataset. We select two model architectures and assess the top-1 accuracy of these models trained on MSWC extractions. Furthermore, we compare these accuracies to the same models trained on the current standard for keyword spotting research, Google's Speech Commands dataset (GSC) [32]. In particular, we seek to determine if a domain gap exists between manually recorded keywords (GSC) and extracted keywords (MSWC), spoken in the context of a full sentence, where coarticulation effects may alter pronunciation.

To investigate this domain gap, we do the following: (1) We cross-compare the test accuracy of two DSCNN models [3], one trained on five keywords chosen from the GSC dataset and one trained on the same five keywords from MSWC and then assess the GSC-trained model's classification performance on MSWC data and vice versa, (2) repeat the first experiment but filter the MSWC data via Sec. 4.6's outlier metric and (3) fine-tune single-target models following [18] using only five randomly chosen samples per keyword.

We select five target keywords from GSC based on the number of samples available for comparison in MSWC. Table 6 shows the total number of samples available in both GSC and MSC for these five keywords. Our dataset reflects word frequencies in natural speech, and consequently exhibits imbalanced classes compared to GSC, a manually collected dataset. We therefore choose *left, right, down, yes,* and *off* as the number of samples for these 5 words in GSC approximately matches

Table 6: GSC targets.

| keyword | # GSC | # MSWC |
|---------|-------|--------|
| left    | 3801  | 5575   |
| right   | 3778  | 7583   |
| down    | 3917  | 8560   |
| yes     | 4044  | 3402   |
| off     | 3745  | 6486   |

---

[2]We maximized the number of anomalous samples found in the farthest 2% percentile in listening tests on 8 English words distinct from the 25 words later used to evaluate our metric's performance.

Table 7: Domain gap between our MSWC extracted keywords and manually recorded Google Speech Commands (GSC). Rows indicate the testing dataset; columns indicate the training dataset.

(a) 5 Target DSCNN models.

|  |  | Train | |
|---|---|---|---|
|  |  | MSWC | GSC |
| Test | MSWC | 85.3% | 60.4% |
|  | GSC | 78.6% | 85.2% |

(b) DSCNN on MSWC inliers.

|  |  | Train | |
|---|---|---|---|
|  |  | MSWC | GSC |
| Test | MSWC | 88.0% | 62.3% |
|  | GSC | 78.4% | 85.2% |

(c) Single Target 5-Shot models.

|  |  | Train | |
|---|---|---|---|
|  |  | MSWC | GSC |
| Test | MSWC | 94.7%±3.9 | 87.6%±7.4 |
|  | GSC | 85.2%±7.4 | 89.0%±3.7 |

our dataset. We use an 80:10:10 ratio for train/val/test splits, following GSC's published splits [32] and MSWC's splits defined in Sec. 4.5. For all evaluations, we use a 49x40 input spectrogram generated by TensorFlow's Microfrontend [30] on 1-sec 16KHz wav encodings of GSC and MSWC clips. We refer to [3] for all other DSCNN hyperparameters and [18] for hyperparameters used in 5-shot transfer learning.

We report baseline results of our top-1 classification accuracies in Tables 7a, 7b, and 7c. In Table 7a, we use the full test split of GSC and MSWC for each keyword, combined with random samples of background audio and unknown samples each equal to the average number of samples per target keyword. Unknown samples are drawn uniformly from non-target English words in GSC and MSWC. We observe a drop in accuracy between a DSCNN model trained and evaluated on GSC data (85.2%) and a model trained on MSWC data and evaluated on the same GSC data (78.6%). The relatively small size of the measured domain gap suggests extracted keywords hold promise for utility in keyword spotting applications. Future work will seek to close this gap via domain adaptation.

Furthermore, we observe potential evidence in Table 7a that our dataset can provide additional robustness. We note that the relatively low accuracy (60.4%) of the 5-target model trained on GSC and tested on MSWC is due primarily to misclassifying target keywords as 'unknown'. This behavior is likely due to the fact that the GSC model is trained only on high quality, manually recorded target samples, and therefore is more likely to classify many samples in a wider distribution as 'unknown'. Models trained on smaller curated KWS datasets may therefore be brittle in practice.

Table 7b reports results for the same setup as Table 7a, after selecting the closest 80% of samples from MSWC using our outlier metric (Sec. 4.6) with $k = 5$ and 50 samples. The accuracy increase for models trained on MSWC and GSC on MSWC test data show that our outlier metric discards anomalous samples from crowdsourced data and boundary errors in forced alignment estimates.

Table 7c aggregates the mean and standard deviation of classification accuracies for 5-shot single-target models for each of the 5 target keywords in Table 6. Each model has three classes: *target keyword, unknown,* and *silence/background-noise*. For each target keyword, we use 5 random seeds where each random seed corresponds to a different selection of five 1-second training samples per target model (hence, Table 7c aggregates 25 models fine-tuned on MSWC target data and 25 on GSC data). Our embedding model is pretrained on our dataset to classify 760 keywords across 9 languages, and fine-tuned on the five training examples as described in [18]. Importantly, the keywords in Table 7c were not observed during pretraining of the embedding model, ensuring our few-shot results are representative for arbitrary keywords. Our results indicate that our dataset can be used to achieve high accuracy KWS models through fine-tuning, and the relatively small number of samples for keywords in our low resource languages (Table 2) is not a barrier to producing accurate KWS models.

## 6 Downstream Applications and Broader Impact

**Motivation, Ethics, and Biases:** We provide useful, free, and open spoken word data in under-resourced languages. Datasets of this nature are instrumental in democratizing speech technology, expanding the inclusivity of research, and widening the reach of voice applications. Here, we consider potential biases in our dataset. For our initial release, MSWC is built exclusively on pre-existing Common Voice data, thus, biases in Common Voice, such as gender or accent bias, can propagate into MSWC. Prior work has explored demographic biases in Common Voice [22], but as MSWC expands to additional datasets, we must be mindful of the biases contained in those corpora. In addition, MSWC may contain biases which stem from forced alignment. Alignments may be of lower quality across low resource languages and for shorter words [9]. Generally speaking, as the size of

the dataset grows, the quality of alignments generated by the Montreal Forced Aligner will increase. Our (optional) outlier metric (Sec. 4.6) may inadvertently introduce additional bias - for example, too few clusters may exclude some accents or non-typical speech. We report estimated errors in English, but leave other languages to future analysis. We provide a Dataset Datasheet [12] in the supplementary material that describes the collection, distribution, and recommended uses of MSWC.

**Applications:** MSWC is in use for Sustainable Development Goals in collaboration with Makerere University. Public sentiment can aid in understanding the impact and reach of public health interventions—in particular, radio call-in shows often reflect sentiment in low-internet connectivity areas. Tools for automated radio monitoring of limited vocabulary terms help healthcare providers and organizations such as the United Nations find relevant discussions to aid in decision support [21].

To aid with COVID-19 response, we have used MSWC to develop a deployable tool for monitoring Ugandan public radio broadcasts for COVID-19 keywords in the Luganda language. As a low-resource language, it is cost-prohibitive to record thousands of keyword samples for each term of interest. Hence, we use MSWC to pre-train a large multilingual embedding model, and fine tune keyword search models using only five to ten example recordings in Luganda. We achieve an average 58% true positive rate across five COVID-19 terms in Luganda in our initial deployment candidate.

KWS datasets are instrumental to the development of many other applications. The data can be used as the target word of a model [4], or to populate the 'other' category of non-target keywords. When targeting low-resource languages, the data can pre-train a model to achieve better accuracy with fewer samples in the target language. Other use cases include wake words for virtual assistants [29], keyword search [20], and voice interfaces for embedded devices [3].

# 7   Long-term Support and Future Work

In order for MSWC to serve as a long-term global resource to all users, we are collaborating with MLCommons (`mlcommons.org`), a non-profit organization that supports machine learning innovation to benefit everyone. We strive to meet four criteria: (1) Relevance - providing updates and expanding the dataset, (2) Accuracy - fixing errors, e.g., erroneous alignments, (3) Usability - offering friendly licensing terms for research and commercial applications, and (4) Accessibility - freely downloadable.

We will keep the dataset updated using our automated and scalable data engineering pipeline (Sec. 4) and generate future versions of MSWC, given that Common Voice performs public releases regularly. Per release, we will make the entire corpus available, including alignments, audio clips, and dataset splits, analogous to our initial release. Each release will bring improvements to the data engineering pipeline, such as improved alignment quality. We support TensorFlow Datasets (TFDS) [31], enabling users to import our dataset in one line of code. We plan to expand the contents of MSWC beyond Common Voice as our system can ingest other audio and transcription sources (Section 4.2).

# 8   Conclusion

Voice interfaces hold promise to democratize access to technology, but the lack of large multilingual datasets has prevented their proliferation. We present a spoken word corpus in 50 languages with over 23 million examples, as a key step in achieving global reach for speech technology. For many languages, our corpus is the first available keyword dataset. We detail our dataset's contents, and report comparable accuracies on models trained using our corpus relative to prior datasets. MSWC is available to download and will be hosted, maintained, and advanced by the MLCommons non-profit organization at `https://mlcommons.org/en/multilingual-spoken-words`.

## Acknowledgements

We acknowledge the help of the individuals who contributed and the organizations that supported this project. We recognize Dr. Joyce Nakatumba-Nabende and Jonathan Mukiibi at Makerere University for their collaboration on our radio monitoring work. We thank Tejas Prabhune from the Evergreen Valley High School in California (USA) for his work on our initial data quality analysis. Sharad Chitlangia is a student of BITS Pilani (India) and did work as an intern at Harvard University. This work was sponsored in part by the Semiconductor Research Corporation (SRC) and Google.

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

# A  Appendix

## 1.1  Word Error Rate Analysis

In Table 8, we provide the full table of word error rates from Section 4.6. To recap, we conduct hearing tests on 25 randomly chosen keywords in English to examine the efficacy of our outlier metric. We list the number of extractions in our dataset (# Clips) for each keyword choose 50 at random to cluster ($k = 5$) in feature-space using our embedding representation. The remaining clips are then expressed as embedding vectors and sorted by minimum distance to each cluster, and the Word Error Rate (WER) is reported as a percentage when listening to the nearest 50 and farthest 50 samples in the sorted dataset.

The word error rates of the nearest 50 samples for each keyword (which have an outlier metric mean $\pm$ stdev value of $1.67 \pm 0.19$) are much smaller than the word error rates of the farthest 50 samples (with a correspondingly larger outlier metric mean and stdev of $3.96 \pm 0.3$). Using the same method, we examine 25 randomly picked words in Spanish and German and find an average WER of 4.6 percent in Spanish and 1.04 percent in German for the nearest 50 samples. In comparison, we estimate an average WER of 21.4 percent in Spanish and 29.6 percent in German for the farthest 50 samples. In Table 9, we show our analysis of our outlier metric's performance on the 25 words each for German and Spanish, following the same approach described in Sec 4.6 and reiterated above.

Table 8: Word error rates (WER) by outlier metric for 25 English words.

| Word | # Clips | Near WER | Far WER |
|---|---|---|---|
| may | 7551 | 4% | 32% |
| did | 14353 | 32% | 32% |
| soon | 3269 | 6% | 34% |
| shirt | 1299 | 0% | 54% |
| style | 1760 | 0% | 12% |
| taken | 1734 | 0% | 12% |
| stood | 1739 | 0% | 22% |
| watch | 1476 | 4% | 28% |
| happy | 2095 | 2% | 8% |
| entire | 1354 | 0% | 12% |
| engine | 1043 | 0% | 18% |
| nature | 1018 | 0% | 12% |
| you've | 1886 | 0% | 26% |
| reading | 1594 | 8% | 58% |
| village | 2801 | 0% | 34% |
| outside | 2075 | 0% | 2% |
| strange | 1720 | 0% | 18% |
| current | 1836 | 10% | 4% |
| musical | 1108 | 0% | 16% |
| followed | 1307 | 0% | 10% |
| learning | 1075 | 2% | 18% |
| provided | 1627 | 0% | 4% |
| difficult | 1771 | 0% | 8% |
| political | 1474 | 0% | 8% |
| performance | 1007 | 6% | 12% |

Table 9: Word error rates (WER) by outlier metric for two additional languages. We observe an average WER of 4.6% for the closest 50 samples in Spanish and 1.04% in German, contrasted with an average WER of 21.4% for the farthest 50 samples in Spanish and 29.6% in German

(a) German

| Word | # Clips | Near WER | Far WER |
|---|---|---|---|
| für | 25024 | 2% | 30% |
| was | 11920 | 4% | 32% |
| erst | 2546 | 4% | 38% |
| geht | 3920 | 0% | 42% |
| ihre | 5285 | 2% | 26% |
| mich | 5309 | 4% | 42% |
| muss | 8656 | 0% | 44% |
| bitte | 4093 | 0% | 32% |
| große | 1221 | 0% | 22% |
| lässt | 2090 | 0% | 24% |
| später | 3026 | 0% | 30% |
| können | 6175 | 0% | 78% |
| niemand | 1073 | 0% | 14% |
| letzten | 1048 | 0% | 44% |
| besteht | 1445 | 0% | 38% |
| gehören | 943 | 0% | 42% |
| zweiten | 1244 | 0% | 26% |
| familie | 1059 | 0% | 18% |
| erhielt | 889 | 0% | 14% |
| zunächst | 1381 | 4% | 16% |
| mitglied | 858 | 2% | 34% |
| gegenüber | 820 | 4% | 26% |
| verwendet | 1127 | 0% | 12% |
| geschichte | 1097 | 0% | 4% |
| universität | 1052 | 0% | 14% |

(b) Spanish

| Word | # Clips | Near WER | Far WER |
|---|---|---|---|
| día | 1110 | 2% | 10% |
| una | 17895 | 32% | 14% |
| ella | 1398 | 8% | 14% |
| pero | 3094 | 4% | 38% |
| están | 1407 | 8% | 52% |
| hasta | 1801 | 14% | 24% |
| nació | 1101 | 0% | 14% |
| algunos | 1252 | 6% | 18% |
| carrera | 1009 | 10% | 20% |
| durante | 2931 | 0% | 22% |
| embargo | 1924 | 0% | 18% |
| estudió | 935 | 0% | 10% |
| familia | 1168 | 0% | 30% |
| general | 958 | 0% | 6% |
| primera | 1803 | 2% | 40% |
| siempre | 958 | 0% | 28% |
| entonces | 1426 | 10% | 26% |
| historia | 1106 | 4% | 20% |
| nacional | 1199 | 8% | 26% |
| encuentra | 2945 | 0% | 22% |
| principal | 1024 | 4% | 16% |
| siguiente | 890 | 0% | 26% |
| diferentes | 879 | 0% | 6% |
| universidad | 1662 | 4% | 8% |
| posteriormente | 1120 | 0% | 26% |

## 1.2 Extended Parts-of-Speech and Semantic Analysis in Additional Languages

We extend our analysis for Parts-of-Speech tagging and semantic characterization from Section 3.4, to provide additional examples in English, Spanish, and Arabic, and to include examples for six additional languages: German, Greek, Russian, Turkish, Vietnamese, and Chinese. Our analysis code is open-sourced, enabling users to explore similar views into the other languages in our dataset. POS tags are generated using the "log-linear part-of-speech tagger" [15] in the multilingual Stanza [24] library. Zero-shot semantic characterization is performed using an NLI model based on RoBerta [7].

| Category | #K | #C | German |
|---|---|---|---|
| Adjective | 15498 | 515439 | hauptstadt, gut, später |
| Verb | 13248 | 666820 | gibt, kommt, bitte |
| Noun | 7805 | 454323 | zeit, stadt, herr |
| Proper noun | 4286 | 178091 | zum, zur, vom |
| Adverb | 999 | 359994 | ein, auch, noch |
| Pronoun | 162 | 451875 | das, sie, sich |
| Auxiliary | 96 | 324801 | ist, sind, hat |
| Other | 69 | 1880 | hei, hallo, via |
| Numeral | 67 | 31045 | zwei, drei, vier |
| Adposition | 44 | 280245 | von, mit, auf |
| Determiner | 41 | 468697 | die, der, den |
| Punctuation | 31 | 1555 | ach, wochen, welch |
| Subordinating Conjunction | 14 | 28779 | dass, wenn, weil |
| Coordinating Conjunction | 10 | 92660 | und, aber, oder |
| Particle | 1 | 42287 | nicht |

| Category | #K | #C | Greek |
|---|---|---|---|
| Noun | 350 | 5125 | βασιλόπουλο, βασιλιάς, στη |
| Verb | 324 | 4355 | είπε, είχε, ρώτησε |
| Adverb | 104 | 3196 | που, στα, γιατί |
| Adjective | 62 | 723 | όλα, όλοι, πρωτομάστορης |
| Pronoun | 58 | 4316 | του, μου, την |
| Numeral | 18 | 592 | ένα, μια, δυο |
| Adposition | 9 | 1353 | στο, από, για |
| Auxiliary | 9 | 673 | είναι, ήταν, έχει |
| Subordinating Conjunction | 6 | 253 | πως, ότι, όταν |
| Coordinating Conjunction | 5 | 2116 | και, ούτε, αλλά |
| Determiner | 3 | 488 | τον, των, ενός |
| Punctuation | 3 | 183 | στην, κυρ, μες |
| Particle | 2 | 514 | δεν, μην |
| Proper noun | 1 | 18 | γερο |
| Other | 1 | 7 | άλφα |

| Category | #K | #C | Russian |
|---|---|---|---|
| Noun | 6966 | 203924 | слово, безопасности, конференции |
| Verb | 4634 | 94091 | является, будет, имеет |
| Adjective | 3485 | 69481 | должны, необходимо, международного |
| Adverb | 437 | 37905 | как, также, еще |
| Determiner | 146 | 21234 | этой, эти, этот |
| Numeral | 86 | 7857 | один, два, три |
| Pronoun | 81 | 42294 | что, это, его |
| Adposition | 27 | 8036 | для, между, без |
| Particle | 16 | 4287 | только, нет, вот |
| Subordinating Conjunction | 8 | 2650 | чтобы, если, когда |
| Interjection | 7 | 579 | при, аль, ага |
| Proper noun | 5 | 137 | firefox, microsoft, mozilla |
| Auxiliary | 4 | 584 | будем, буду, будучи |
| Coordinating Conjunction | 4 | 1104 | или, либо, причем |

| Category | #K | #C | Turkish |
|---|---|---|---|
| Noun | 1567 | 27441 | yıl, avro, devam |
| Verb | 773 | 15161 | var, değil, oldu |
| Adjective | 324 | 8624 | yeni, nasıl, büyük |
| Proper noun | 114 | 2311 | kosova, türkiye, sırbistan |
| Adverb | 83 | 8505 | bir, çok, daha |
| Pronoun | 77 | 2772 | bunun, bunu, buna |
| Numeral | 36 | 6318 | iki, bin, beş |
| Punctuation | 21 | 1190 | yüz, kırk, nedir |
| Adposition | 15 | 1765 | için, kadar, olarak |
| Coordinating Conjunction | 12 | 1712 | ancak, fakat, ile |
| NaN | 10 | 113 | nelerdir, zamandır, vardır |
| Auxiliary | 9 | 721 | ise, musunuz, misiniz |
| Determiner | 7 | 826 | her, bazı, tüm |
| Interjection | 1 | 23 | hey |

| Category | #K | #C | Vietnamese |
|---|---|---|---|
| Noun | 15 | 118 | con, người, ông |
| Other | 7 | 78 | không, được, còn |
| Proper noun | 6 | 32 | đâu, đây, vậy |
| Verb | 6 | 50 | nói, tao, nhìn |
| Coordinating Conjunction | 5 | 48 | thì, rồi, như |
| Adposition | 4 | 33 | của, với, cho |
| Particle | 3 | 16 | mày, thật, thôi |
| Numeral | 1 | 11 | một |

| Category | #K | #C | Chinese (zh-CN) |
|---|---|---|---|
| Proper noun | 37 | 884 | 奥地利, 巴伐利亚州, 马来西亚 |
| Noun | 34 | 525 | 俱乐部, 美术馆, 篮球队 |
| Numeral | 10 | 109 | 十八年, 四十四, 三尖杉 |
| Verb | 6 | 52 | 影响力, 意味着, 发源地 |
| Particle | 3 | 27 | 大部份, 的町名, 副作用 |
| Adjective | 1 | 13 | 连续剧 |
| Adverb | 1 | 5 | 日常生活 |

Figure 4: Additional Parts-Of-Speech Tags in German, Greek, Russian, Turkish, Vietnamese, and Chinese with representative samples. Number of keywords (#K) and clips (#C). Sorted by #K.

We expand on Table 4 and provide additional data on semantic classification in English, Spanish, and Arabic in Table 10, in reference to our discussion in Sec. 3.4. In Fig. 4, we provide sample data on POS tags in German, Greek, Russian, Turkish, Vietnamese, and Chinese (zh-CN language code). Additionally, in Fig. 5, we provide sample data on semantic keyword characterization for these six additional languages, following the same approach as in Sec. 3.4.

Table 10: An expanded view of semantic keyword characterization in English, Spanish and Arabic via zero-shot multilingual NLI with representative samples. Number of clips (#C) and keywords (#K). Sorted by #K in English.

| Category | # C | # K | English | #C | #K | Spanish | #C | #K | Arabic |
|---|---|---|---|---|---|---|---|---|---|
| Event | 3M | 1K | School, Preakness, Commanded | 153K | 238 | Campeonato, Episodio, Eurovisión | 2K | 30 | يقال، الصحل يحبها |
| Human activity | 1M | 820 | Shooting, Prefers, Attacking | 78K | 211 | Expandiendo, Visitando, Carpintería | 1K | 28 | ستغادر، المشترين، لدغني |
| Location | 391K | 379 | Home, County, Desert | 213K | 324 | Zona, Pueblo, Marítima | 333 | 14 | يقال، الصحل يحبها |
| Name | 327K | 303 | Margot, Cooney, Alvin | 30K | 128 | Eduardo, Peter, Francisco | 1K | 24 | انا ساعبقي ادري |
| Animal | 131K | 281 | Sheep, Camel, Muzzle | 99K | 137 | Águila, Especies, Jaguar | 1K | 25 | فيه تفاحة باللون |
| Number | 301K | 279 | Second, Six, Stringent, | 31K | 75 | Multiplicar, Primero, Estadísticos | 493 | 17 | عشرة، واحد ثلاثة |
| General reference | 113K | 220 | Understand, Generally, Trivial | 216K | 319 | Recomendaciones, Recompensa, Frecuencias | 185 | 8 | طريقة القانون، التعب |
| Common words | 218K | 217 | Often, Popular, Usually | 56K | 172 | Tambien, Ademas, Biográfico | 206 | 7 | عديدة، تمطر مخطئ |
| City | 106K | 212 | York, London, California | 34K | 94 | Madrid, Berlín, Oxford | 49 | 4 | ايطاليا، بيكاسو البلدة |
| Technology | 52K | 191 | Gramophone, Television, Videotape | 21K | 106 | Automotriz, Lego, Sensor | 92 | 5 | التلفاز، السيارات، الاخير |
| Culture | 52K | 180 | Popularize, Music, Style | 15K | 86 | Tradicional, Concertista, Cantante | 57 | 4 | الصينية بمهارة، فوق |
| Language | 92K | 132 | Arabic, English, Words | 8K | 69 | Español, Lengua, Inglesa | 98 | 7 | اللغة بالانجليزية يابان |
| Game | 48K | 131 | Play, Kirby's, Football | 8K | 46 | Deportes, Partido, Equipo | - | - | - |
| Political | 37K | 124 | Impeached, Campaigns, Democrats | 13K | 86 | Intendentes, Libertario, Poder | 2K | 32 | رؤية المفضل معي |
| History | 52K | 119 | Stories, Ancient, Roman | 12K | 73 | Conquistas, Emperador, Históricas | 257 | 8 | قبل ائتذكر عمرها |

| Category | #K | #C | German |
|---|---|---|---|
| political | 756 | 2216421 | die, der, ist |
| general reference | 268 | 144785 | nur, immer, nichts |
| human activity | 264 | 221804 | ich, wir, man |
| common words | 258 | 168175 | wie, wieder, habe |
| animal | 247 | 212687 | den, einen, einer |
| location | 234 | 104881 | unter, liegt, steht |
| society | 212 | 111054 | hat, alles, anderen |
| event | 197 | 108376 | wird, werden, kommt |
| city | 186 | 66360 | hauptstadt, stadt, gebäude |
| health | 184 | 90310 | haben, sein, hast |
| name | 172 | 52039 | mein, meine, heißt |
| technology | 162 | 53924 | können, neue, neuen |
| social science | 155 | 40135 | zwischen, kurz, familie |
| science | 149 | 34182 | weiß, wissen, universität |
| number | 138 | 94758 | eine, einem, zwei |
| history | 126 | 31221 | einmal, jahren, jahre |
| language | 123 | 27136 | deutschen, lang, deutsche |
| geography | 106 | 18763 | land, gebiet, insel |
| art | 104 | 19830 | art, schön, werke |
| engineer | 100 | 27568 | kann, fährt, kühlschrank |
| religion | 100 | 17252 | vater, kirche, glaube |
| culture | 92 | 15216 | europäische, musik, hören |
| game | 69 | 12931 | kinder, spielen, spielt |
| mathematic | 39 | 3193 | mag, mach, zylinder |
| software | 33 | 3134 | firefox, software, anwendung |
| philosophy | 21 | 1133 | philosophie, philippinen, pfui |

| Category | #K | #C | Greek |
|---|---|---|---|
| human activity | 60 | 8145 | μου, της, σου |
| common words | 40 | 4337 | και, είχε, πως |
| location | 39 | 3017 | που, στο, στα |
| political | 34 | 4701 | του, δεν, από |
| number | 18 | 1147 | είναι, ένα, μια |
| animal | 17 | 687 | βασιλόπουλο, παρά, χαράβια |
| name | 14 | 820 | τον, κουλός, ποιος |
| general reference | 11 | 259 | τίποτα, μην, κανένας |
| society | 8 | 118 | κανείς, άλλες, σιωπηλά |
| event | 8 | 213 | φώναξε, πες, πει |
| health | 8 | 134 | καλά, γερο, καρδιά |
| science | 5 | 66 | γνώση, ξέρει, μάθω |
| technology | 5 | 58 | χειράμαξα, νέο, καινούρια |
| religion | 4 | 57 | πολύδωρος, θείος, θείου |
| language | 3 | 38 | λόγια, μάθει, λέξεις |
| game | 2 | 16 | χουτσός, κονσόλα |
| history | 2 | 19 | βιβλίο, διηγήθηκε, παλιά |
| city | 2 | 37 | μεγάλο, μεγάλη, σχολείο |
| social science | 2 | 19 | σκοπό, νόημα |
| art | 1 | 5 | σκηνή |
| mathematic | 1 | 28 | αλυσίδα, μάθημα |
| geography | 1 | 5 | σύνορα |
| philosophy | 1 | 9 | σοφά |
| software | 1 | 6 | πρόγραμμα |

| Category | #K | #C | Russian |
|---|---|---|---|
| political | 290 | 169316 | что, это, для |
| location | 183 | 65472 | нас, хотел, время |
| number | 143 | 39045 | имеет, уже, более |
| human activity | 141 | 37049 | усилия, человека, роль |
| society | 137 | 32908 | наций, организации, всех |
| name | 112 | 29457 | его, слово, является |
| event | 96 | 21131 | будет, конференции, было |
| health | 86 | 17181 | безопасности, быть, необходимо |
| social science | 78 | 10953 | без, процесс, характер |
| animal | 73 | 14473 | кроме, такие, просто |
| technology | 70 | 10328 | ядерного, оружия, качестве |
| common words | 63 | 13958 | как, также, все |
| general reference | 57 | 5338 | связи, сотрудничество, вместе |
| history | 50 | 4595 | году, года, год |
| mathematic | 46 | 3505 | если, задача, задачи |
| science | 46 | 3784 | доклада, деле, доклад |
| art | 35 | 2978 | международной, выразить, создать |
| language | 34 | 2130 | речь, говорить, высказать |
| software | 33 | 2719 | проекта, программы, обеспечения |
| culture | 32 | 2056 | европейский, носят, корейской |
| city | 31 | 2009 | экономического, экономики, столица |
| engineer | 25 | 938 | опыт, опытом, подготовки |
| religion | 21 | 968 | при, исламской, верим |
| game | 19 | 854 | играет, играть, роли |
| geography | 12 | 335 | линии, широкие, континента |
| philosophy | 12 | 302 | принципы, мудрость, мудрости |

| Category | #K | #C | Turkish |
|---|---|---|---|
| political | 103 | 23785 | ancak, bin, yüz |
| human activity | 68 | 9716 | kişi, katıldı, ediyor |
| common words | 66 | 7137 | her, nasıl, avro |
| number | 62 | 14283 | bir, iki, çok |
| location | 52 | 4486 | yer, ülke, ülkeden |
| name | 48 | 4089 | ise, sona, ilk |
| society | 38 | 2887 | durum, siyasi, durumda |
| history | 33 | 1759 | yıl, zaman, geri |
| general reference | 32 | 2006 | yaklaşık, sadece, bunlar |
| animal | 30 | 1948 | bile, hala, reddetti |
| health | 22 | 1232 | iyi, ayında, musunuz |
| technology | 20 | 1082 | yeni, proje, gelecek |
| event | 18 | 995 | olacak, geldi, festival |
| geography | 16 | 784 | büyük, civarında, dünya |
| game | 12 | 251 | maç, çocuklar, maçı |
| science | 12 | 213 | bilgi, uzmanlar, tesis |
| culture | 7 | 234 | türk, etnik, kültür |
| religion | 7 | 126 | maliyeti, dini, bedeli |
| software | 6 | 159 | projenin, programın, program |
| language | 6 | 65 | küçük, dil, bölünmüş |
| art | 5 | 116 | arttı, sanatçı, sanatçının |
| engineer | 5 | 46 | uzman, sektörde, memur |
| social science | 3 | 31 | çeşitlilik, kimseyi, eğitimi |
| city | 3 | 36 | kent, kentin, kente |
| mathematic | 1 | 8 | çözmek |
| philosophy | 1 | 8 | düşünceye |

| Category | #K | #C | Chinese (zh-CN) |
|---|---|---|---|
| location | 18 | 760 | 奥地利, 巴伐利亚州, 马来西亚 |
| number | 10 | 128 | 十八年, 四十四, 二十一 |
| human activity | 8 | 236 | 俱乐部, 事务所, 影响力 |
| name | 6 | 87 | 莎草科, 虎耳草科, 戈亚斯 |
| animal | 5 | 64 | 十字花科, 鹅观草, 龙胆科 |
| city | 4 | 63 | 维也纳, 大阪府, 巴拿马 |
| health | 3 | 29 | 玄参科, 蛋白质, 珍珠菜 |
| history | 3 | 35 | 巡洋舰, 记忆体, 安土桃山时代 |
| culture | 3 | 26 | 连续剧, 哥特式, 主题曲 |
| political | 3 | 24 | 评论家, 主任委员, 反对派 |
| event | 2 | 8 | 获得者, 副作用 |
| science | 2 | 14 | 天文学家, 科进士 |
| technology | 2 | 31 | 航空母舰, 机器人 |
| game | 1 | 30 | 篮球队 |
| art | 1 | 31 | 美术馆 |
| mathematic | 1 | 9 | 大学部 |
| philosophy | 1 | 21 | 哲学家 |
| religion | 1 | 13 | 东正教 |
| society | 1 | 6 | 人类学 |

| Category | #K | #C | Vietnamese |
|---|---|---|---|
| human activity | 10 | 294 | không, thì, của |
| number | 3 | 33 | con, một, sao |
| event | 2 | 13 | chết, chuyện |
| location | 2 | 16 | nhà, đâu |
| animal | 1 | 7 | cái |
| language | 1 | 15 | nói |
| religion | 1 | 8 | trinh |

Figure 5: Additional semantic keyword characterization in German, Greek, Russian, Turkish, Vietnamese, and Chinese via zero-shot multilingual NLI with representative samples. Number of keywords (#K) and clips (#C). Sorted by #K.

