# OpenReview forum: "Multilingual Spoken Words Corpus"
_NeurIPS.cc/2021/Track/Datasets_and_Benchmarks/Round2 — NeurIPS 2021 Datasets and Benchmarks Track (Round 2)_

### Official Review · Reviewer_RhXJ · 2021-09-08
**A Large-Scale Multilingual Spoken Words Corpus for Keyword Spotting**

**Rating:** 7
**Confidence:** 3
**Correctness:** The data collection and evaluation me…
**Clarity:** This paper is well written and easy t…

**Strengths:**

- This multilingual keyword dataset will be a very useful resource for building keyword spotting models, especially for low resource languages.
- Authors have done a good job to analyze the features of the proposed MSWC, including the distribution of the number of samples across all 50 languages, keyword length variation, and keyword characterization.
- The experimental results demonstrate the effectiveness of using MSWC as a pre-training dataset for building keyword spotting models.

**Weaknesses:**

The authors mentioned that previous works [1,2] have applied a similar method to extract keyword spotting data. What is the difference between their dataset and MSWC?

[1] Query-by-example keyword spotting system using multi-head attention and soft-triple loss. ICASSP 2021
[2] Reddi. Few-shot keyword spotting in any language. Interspeech 2021

**Additional Feedback:**

None

**Documentation:**

This project has a clear maintenance plan. Overall, this dataset is well documented.

**Ethics:**

The authors have included an ethical statement at the end of the paper. I do not have any additional concerns.

**Relation To Prior Work:**

See weakness.

**Summary And Contributions:**

This paper introduces the Multilingual Spoken Words Corpus (MSWC) which contains more than 340,000 keywords, totaling 23.7 million 1-second spoken examples. MSWC is extracted from Common Voice and automatically annotated by applying a forced alignment tool (Montreal Forced Aligner). In addition, the authors apply k-means clustering and nearest-neighbors strategies to automatically identify anomalous samples. Experimental results show that a keyword spotting model pre-trained on MSWC can achieve strong zero-shot and few-shot performance in manually recorded Google’s Speech Commands dataset.

---

### Official Review · Reviewer_cwfU · 2021-09-13
**Useful multilingual resource / more experiments could have been included**

**Rating:** 7
**Confidence:** 4
**Correctness:** The dataset is constructed in a sound…
**Clarity:** The paper is clear and well written.

**Strengths:**

The dataset addresses a gap in the literature as it is the first to contain spoken word data for such a large range of languages, including 26 low-resource languages and 12 medium-resource ones, for many of which this is their first publicly available spoken word dataset. This is a significant step forward.
The dataset and source code (that includes the keyword extraction pipeline) are open-source.
The paper provides detailed analyses of the data together with experiments that quantify the domain gap between automatically extracted and manually recorded keywords.


**Weaknesses:**

To assess the efficacy of the outlier metric, the authors conduct experiments only on English and not on any other of the 49 languages the dataset contains. Similarly for the experiments on quantifying the domain gap. There are some analyses but for a limited set of languages (English, Spanish, Arabic). The authors could have included a larger range of diverse languages.

**Additional Feedback:**

Section 4.4 -> what are the different heuristics and thresholds that are applied? Only one is mentioned here.

Section 5, experimental setting 1: how many samples per keyword are used here?

"we empirically select k=5 as the number of clusters" -> how was this selected?

lines 167--168: please elaborate and expand more on this as it is a key part of the pipeline

Perhaps you can swap Sections 3 and 4 and/or present the construction of the dataset first and then continue with the data analyses.

The following need to be fixed:
line 124: "we contain"
line 135: "a give languages"
line 135: "broadly with"
line 157: "to grow to include"

Subject--verb agreement errors in caption of Table 6.

line 158: "audio, and" -> remove comma
line 212: "lowest" -> smallest
line 253: "and vice versa" is not quite right here


**Documentation:**

There is sufficient detail on data collection and organization, availability and maintenance.

**Ethics:**

The authors provide a datasheet in the supplementary material. They mention that "to our best efforts we have ensured that the dataset does not contain any data that might be considered sensitive in any way". Could you please elaborate more on that? How have you ensured that?

**Relation To Prior Work:**

Differences to previous work are clearly discussed.

**Summary And Contributions:**

[Thank you for your response.]

The paper presents MSWC, a new multilingual corpus of spoken words consisting of audio data in 50 languages. The dataset is quite large and contains over 340K keywords and over 6k hours of spoken examples from around 115K source speakers. MSWC is generated automatically by applying forced alignment on Common Voice, an existing crowdsourced speech corpus. To address potential issues on the quality of produced alignments, the authors further propose an 'outlier' metric based on nearest-neighbors to automatically estimate the quality of extracted keywords.

---

### Official Review · Reviewer_PxiV · 2021-09-20
**A Dataset of 340,000 spoken words from 115,000 speaker in 50 languages**

**Rating:** 7
**Confidence:** 4

**Strengths:**

The dataset contains a large amount of data over a wide range of keywords and languages, particularly low resource languages.
Since the dataset used automated tools to extract audio clips of words from full sentences their artifacts of the process can be present in the dataset, but models trained on MSWC performed similarly with GSC, a manually curated dataset.


**Weaknesses:**

It was mentioned in the Section 6, but it would be interesting to see the quality of the extractions and outlier detentions evaluated for other languages, particularly some of the low resource languages.


**Additional Feedback:**

 Are the demographics for each client_id part of the dataset if the user provided them?


**Clarity:**

The paper was easy to understand.


**Correctness:**

The dataset is well constructed. Since there is a large amount of data an automatic metric is presented that can evaluate and be used to remove audio clips with eros, so there are not hard guarantees that the data is perfect.


**Documentation:**

The data extraction process was described clearly, and the data should be simple to process. The paper also reports the data’s key features and statistics, part of speech, word length, and semantic type.


**Ethics:**

The potential for bias in the dataset, forced alignment, and outlier detection are all discussed in the paper.


**Relation To Prior Work:**

The paper references previously collected keyword spotting dataset, which were much more limited in breadth than the one proposed in the paper.


**Summary And Contributions:**

This paper presents a dataset of 23.7 million 1 second audio clips of 340,000 unique keywords from 115,000 speakers over 50 different languages. The dataset was collected by applying forced alignment to the Common Voice speech corpus to extract individual words, and an outlier metric was developed to detect and remove potentially bad audio clips, at the discretion of dataset users. The dataset is significantly larger in the number of words contained and in the number of languages covered compared to existing datasets.

---

### Official Review · Reviewer_GLKk · 2021-09-20

**Rating:** 8
**Confidence:** 4
**Correctness:** All good.
**Clarity:** Very clear.

**Strengths:**

This should be an extremely useful resource. MSWC dominates every other keyword spotting dataset in size and coverage.

"we include the generated Praat textgrid alignments as part of our dataset release" --> this is excellent

**Weaknesses:**

"The high-resource languages are collectively spoken by 2.75 billion people." what does this mean? This is surely not the actual number of speakers

Someone might want to predict, for an entire _sentence_, whether the sentence contains the keyword or not---do the textgrids cover the entire sentence? or just the keyword portion? Alternately, you could list the source file from Common Voice alongside training examples for those who want to do this

"We train forced alignment from a flat start only on the Common Voice data itself, i.e., we do not rely on any external acoustic models" --> even for low-resource langs? do you use phoneme-based lexica?

"Chinese" = is this CV's "zh-tw", "zh-cn", "zh-hk" or all three?

**Additional Feedback:**

n/a

**Documentation:**

Looks great.

**Relation To Prior Work:**

Extensive review and comparison with existing keyword spotting resources.

**Summary And Contributions:**

This paper introduces the Multilingual Spoken Words Corpus (MSWC), a large dataset of spoken words extracted from the Common Voice dataset via forced alignment. In addition to the cleaning done by Common Voice volunteers, the authors use an anomaly detection approach to find bad examples and supply their code for this to users. Models trained on MSWC are also found to have fairly good performance on the out-of-domain Google Speech Commands.

---

### Decision · Program_Chairs · 2021-10-09

**Decision:**

Accept

**Comment:**

The authors present MSWC, a large dataset of spoken words across 50 languages extracted from Common Voice dataset. The reviewers agree that this dataset is a significant contribution since it is well constructed and larger in terms of number of words and languages compared to existing resources. I recommend acceptance.